# Sex Differences in the Prevalence of Head and Neck Cancers: A 10-Year Follow-Up Study of 10 Million Healthy People

**DOI:** 10.3390/cancers14102521

**Published:** 2022-05-20

**Authors:** Jun-Ook Park, Inn-Chul Nam, Choung-Soo Kim, Sung-Joon Park, Dong-Hyun Lee, Hyun-Bum Kim, Kyung-Do Han, Young-Hoon Joo

**Affiliations:** 1Department of Otolaryngology-Head and Neck Surgery, College of Medicine, The Catholic University of Korea, Seoul 06591, Korea; junook2000@catholic.ac.kr (J.-O.P.); entnam@catholic.ac.kr (I.-C.N.); drchoung@catholic.ac.kr (C.-S.K.); cjidea@naver.com (D.-H.L.); goldgold11@hanmail.net (H.-B.K.); 2Department of Otorhinolaryngology-Head and Neck Surgery, Chung-Ang University College of Medicine, Gwangmyeong Hospital, Gwangmyeon-si 14353, Korea; hypocratis@gmail.com; 3Department of Statistics and Actuarial Science, Soongsil University, Seoul 06978, Korea; hkd917@naver.com

**Keywords:** head and neck neoplasms, squamous cell carcinoma of head and neck, sex characteristics, gender difference, sex difference, alcohol drinking, smoking, cohort studies

## Abstract

**Simple Summary:**

Males are much more susceptible to head and neck cancers than females regardless of whether they drink alcohol or smoke tobacco. Sex differences in the incidence of head and neck cancer are most evident in the lower part of the upper aerodigestive tract. Our results suggest a direction for future research on head and neck cancer epidemiology.

**Abstract:**

Background: Descriptive epidemiologists have repeatedly reported that males are more susceptible to head and neck cancers. However, most published data are those of cross-sectional studies, and no population-based cohort study has yet been published. The aim of this study was to compare the prevalence of head and neck cancers in healthy males with females. Methods: A retrospective cohort study using the Korean National Health Insurance Service database on 9,598,085 individuals who underwent regular health checkups from 1 January to 31 December 2009. We sought head and neck cancers developed during the 10-year follow-up. Results: A total of 10,732 (incidence rate (IR) per 1000 person-years 0.25) individuals were newly diagnosed with head and neck cancer among the 9,598,085 individuals during the 10-year follow-up. The IR was 0.19 in males (8500 affected) and 0.06 in females (2232 affected). Notably, the male–female ratio increased with age below 70 years but decreased thereafter. The male–female difference was most apparent for laryngeal cancer; the male IR was 11-fold higher in the 40 s and 20-fold higher in the 60 s, followed by hypopharyngeal cancer (6.8- and 24.2-fold). Males smoked more and drank more alcohol than females (*p* < 0.0001 *, *p* < 0.0001 *). When never-smokers/-drinkers (only) were compared, males remained at a 2.9-fold higher risk of head and neck cancer than females. The hazard ratios for head and neck cancers in males tended to increase in the lower part of the upper aerodigestive tract: larynx (13.9) > hypopharynx (10.9) > oropharynx (4.4) > nasopharynx (2.9) > sinonasal region (1.8) > oral (1.6). Only the salivary gland cancer incidence did not differ between the sexes; the gland is not in the upper aerodigestive tract. Conclusion: Males are much more susceptible to head and neck cancers than females regardless of whether they drink alcohol or smoke tobacco. Sex differences in the incidence of head and neck cancer are most evident in the 60 s in the lower part of the upper aerodigestive tract, such as the larynx and hypopharynx.

## 1. Introduction

Descriptive epidemiologists have repeatedly reported differences in the cancer prevalence between males and females; males are more susceptible to most cancers [1,2]. The latest report (2019) also found that cancers were 1.2-fold more common in males [3,4]. This may reflect endogenous sex-specific biological differences or exogenous environmental variations that affect health care, cancer perceptions, and treatment compliance [5,6]. Head and neck cancer is the seventh most common cancer worldwide; there were more than 900,000 new cases in 2020 [7]. Most of such cancers are squamous cell carcinomas and include sinonasal, oral, nasopharyngeal, oropharyngeal, hypopharyngeal, laryngeal, and salivary gland cancers [7,8]. The incidence of head and neck cancers in males was higher than in females [3,4]. An analysis of data extracted from the November 2007 Surveillance, Epidemiology, and End Results (SEER) cancer registry revealed that laryngeal (male–female ratio 5.17), hypopharyngeal (4.13), tonsil (3.07), and oropharyngeal (3.06) cancers were among the top 10 cancers with the largest male–female ratios (the tonsils were distinguished from the oropharynx) [1]. The trends in head and neck cancers in South Korea (1999–2012) were similar: the male-to-female ratio exceeded 1.0 for all such cancers; the incidence of oral cancer was 2.8-fold higher in males than in females, and the incidence of pharyngeal and laryngeal cancers was 4.5-fold higher [9]. However, most published data are those of cross-sectional studies using information from cancer registries such as SEER [1,2,3,4,9,10]; no population-based cohort study that followed-up healthy individuals in terms of head and neck cancer incidence has yet been published in English. Thus, we performed a cohort study using the Korean National Health Insurance Service (KNHIS) database on 10 million healthy individuals who underwent regular health checkups in 2009 and who were then followed up for 10 years. We investigated the prevalence of, and risk factors for, head and neck cancers to suggest promising directions for future research.

## 2. Materials and Methods

### 2.1. Data Source and Study Population

The KNHIS is a mandatory public health insurance system administered by the Korean government; it covers about 97% of the entire Korean population (with the exception of the 3% of Medicaid beneficiaries) [11]. The KNHIS provides health care benefits and regular health checkup benefits every 2 years to all subscribers. The KNHIS database contains health information on most Koreans including: demographics; claims (general information on specifications; diagnoses as defined by the International Classification of Disease, Tenth Revision, Clinical Modification (ICD-10-CM) codes); prescriptions and interventions; death information; and regular health check-up data [12].

### 2.2. Study Design and Patient Selection

We performed a retrospective cohort study using the KNHIS database on 10,585,852 individuals who underwent regular health checkups from 1 January to 31 December 2009. Of these, data for 746,403 were unavailable, and we excluded 153,456 with a history of any type of cancer before and within 1 year after the index date, and 87,908 who underwent less than 1 year of follow-up. A total of 9,598,085 were finally included. We sought head and neck cancers developed during the 10-year follow-up by searching the relevant ICD-10-CM codes from 1 January 2010 to 31 December 2018 (Appendix A). The flow chart of subject selection from the KNHIS database is summarized in Figure 1. The following data were extracted from the database: age (years), smoking and alcohol consumption status, income, regular exercise status, height, weight, body mass index, waist circumference, abdominal obesity, glucose level, diabetes mellitus status, blood pressure, hypertension status, estimated glomerular filtration rate, chronic kidney disease status, cholesterol and triglyceride levels, and dyslipidemia status. The study protocol was approved by the Institutional Review Board of the Catholic University of Korea (no. PC21ZISI0235).

### 2.3. Potential Confounders

Variables that might possibly affect the head and neck cancer incidence were regarded as confounders: age; body mass index; smoking status; alcohol consumption level; low income; regular exercise, diabetes mellitus, and hypertension status [13,14,15,16,17,18,19]. Age was classified into <40, 40–64, and ≥65 years. The body mass index was the weight/height squared (kg/m^2^) and was divided into <25 and ≥25 kg/m^2^. Smoking status was classified as “never”, “past”, and “current” as recorded on the baseline survey. Alcohol consumption was based on that per week (“none”; “mild” < 30 g/day; or “heavy” ≥ 30 g/day). Income was determined by the monthly insurance premium (which reflects income). Regular exercise was defined as moderate to vigorous physical activity on at least 4 days a week.

### 2.4. Outcome Measurements

The incidence of head and neck cancers was determined by reference to the ICD-10-CM codes registered during the 10-year follow-up (from 1 year after baseline screening in 2009 until the end of follow-up or 31 December 2018); head and neck cancer was included in a co-payment reduction program for those with critical illnesses. In South Korea, virtually all people apply for such assistance if they are diagnosed with cancer because KNHIS covers 95% of all payments.

### 2.5. Statistical Analysis

All analyses were performed with the aid of SAS ver. 9.4 (SAS Institute, Cary, NC, USA); the *p*-values are two-sided. A *p*-value < 0.05 was considered significant. Descriptive statistics were used to record baseline characteristics. Males and females were compared using the χ^2^ test and one-way analysis of variance. Multivariate analyses employing a Cox’s proportional-hazards model were performed to determine the hazard ratios (HRs) for head and neck cancer incidence by sex. Covariates were not adjusted in regression model 1; age, body mass index, smoking status, alcohol consumption level, low income, and regular exercise status were adjusted in model 2; diabetes mellitus and hypertension status were additionally adjusted in model 3.

## 3. Results

The baseline characteristics of the 9,598,085 subjects are summarized in Table 1. The mean age was 45.63 ± 13.43 years in males and 48.64 ± 14.5 years in females, thus significantly different (*p* < 0.0001 *). Of males, 30.88% stated that they were never-smokers, 24.06% were ex-smokers, and 45.06% were current smokers; among females, 95.04% were never-smokers. Of males, 13.4% were heavy alcohol consumers (≥30 g/day), 54.19% mild consumers (<30 g/day), and 32.07% were non-consumers. Of females, 74.9% were non-consumers, 24.3% were mild consumers, and 1.11% were heavy consumers. Significantly fewer males than females reported that they were never-smokers/-drinkers (both *p* < 0.0001 *). Males exhibited a significantly higher body mass index, a higher income, more regular exercise, and higher rates of diabetes mellitus and hypertension than females (all *p* < 0.0001 *). A total of 10,732 (incidence rate (IR) per 1000 person-years 0.25) individuals were newly diagnosed with head and neck cancer among the 9,598,085 individuals who underwent regular health checkups during the 10-year follow-up. The IR was 0.19 in males (8500 affected) and 0.06 in females (2232 affected). Laryngeal cancer was the most common male cancer and oral cancer the most common female cancer; the IRs for the various cancers are shown in Figure 1. Kaplan–Meier survival curves revealed that males had a higher incidence than females of all new head and neck cancers except salivary gland cancer (Figure 2). The adjusted HR for all head and neck cancers was significantly higher in males than females after adjusting for confounders (HR 2.816 [2.656, 2.985], *p* < 0.0001 *) in Model 3 (Table 2). The adjusted HRs for laryngeal cancer (10.981 [9.171, 13.148], *p* < 0.0001 *), sinonasal cancer (1.758 [1.388, 2.226], *p* < 0.0001 *), hypopharyngeal cancer (9.949 [7.476, 13.239], *p* < 0.0001 *), oropharyngeal cancer (4.589 [3.942, 5.343], *p* < 0.0001 *), oral cancer (1.472 [1.31, 1.654)], *p* < 0.0001 *), and nasopharyngeal cancer (2.805 [2.359, 3.336], *p* < 0.0001 *) were significantly higher in males, but the difference was not significant for salivary gland cancer (HR 1.139 [0.977, 1.327], *p* = 0.0966) as revealed by Model 3. The IRs of head and neck cancers in previously healthy males and females by age are shown in Figure 3 (the details are in Appendix A). The male–female ratios of cancer incidence were 1.0 in the 20 s, 1.5 in the 30 s, 2.5 in the 40 s, 4.4 in the 50 s, 5.4 in the 60 s, and 4.6 in the 70 s. The male–female ratio of head and neck cancers among never-smokers/-drinkers (only) were: hypopharyngeal (9.949), laryngeal (10.981), oral (1.472), sinonasal (1.758), nasopharyngeal (2.805), and oropharyngeal (4.589).

## 4. Discussion

The incidence of cancers varies by sex/gender; these may influence the cancer risk in different ways. Sex is a biological concept determined by the sex chromosomes [20,21,22,23,24]; gender refers to social concepts that can vary by society and time [5,6]. The complex and dynamic interactions between sex and gender throughout the lifespan may affect health care; cancer susceptibility, perception, progression; and treatment compliance. Cancer-preventing lifestyles and minimal exposure to risk factors are critical. Men are more likely to exhibit behavioral risk factors such as alcohol consumption and smoking. However, even after adjusting for these factors, men still evidenced a higher incidence of cancer [2,25,26]. Between-sex differences in anatomy, physiology, body composition, and drug metabolism might affect cancer incidence [27]. Sex hormones may play critical roles in carcinogenesis and cancer susceptibility: androgens are significantly associated with a higher cancer prevalence (and poor outcomes) in men; estrogens seem to exert protective effects in females [20,21,22,23]. However, as the IRs of most cancers are higher even in male children and adolescents, the sex hormones do not fully explain the sex disparity [28]. It has been suggested that sexual differences in the humoral and cell-mediated immune responses to viral infection may explain the differences in cancer prevalence [29,30,31,32]. Generally, females mount stronger humoral and cell-mediated immune responses than men throughout life [33]. Differences in the regulation of the immune response and the infection burden may contribute to sex differences in cancer incidence including human papillomavirus (HPV)-associated cancer.

The head and neck cancer IR in Korean males was about three-fold that in females; we studied a cohort of 10 million. Notably, the male–female ratio increased with age below 70 years but decreased thereafter. The male–female difference was most apparent for laryngeal cancer; the male IR was 11-fold higher in the 40 s and 20-fold higher in the 60 s, followed by hypopharyngeal cancer (6.8- and 24.2-fold). A decreased sexual gap in terms of socioeconomic activity, less exposure to tobacco and alcohol, and reduced sex hormones at ages > 70 years may play roles, but much remains unclear [2,20,21,22,23,25,26,34,35].

Males smoked more and drank more alcohol than females; both are major risk factors for head and neck cancers [36,37,38,39]. Lewin et al. reported that the relative risk (RR) of head and neck cancer among current smokers was 6.5-fold than that of non-smokers. Twenty years of smoking cessation is required before the risk declines to the level of the non-smoker. The RR associated with alcohol consumption ≥ 50 g/day was 5.5 compared to consumption of <10 g/day [36]. In terms of bodyweight management, interestingly, more males (19.7%) than females (15.4%) stated that they exercised regularly, but males were more likely to be overweight (males 37.2% vs. females 27.1%). Diabetes and hypertension rates were higher in males, and the chronic kidney disease and dyslipidemia rates were higher in females. More females than males had low incomes (males 15.0%, females 24.9%). Males remained at a 2.8-fold higher risk of head and neck cancer than females even after adjusting for these confounding factors [13,14,15,16,17,18,19]: the laryngeal cancer risk was 10.9-fold higher and that of hypopharyngeal cancer was 9.9-fold higher in males than females. When never-smokers/-drinkers (only) were compared, males remained at a 2.9-fold higher risk of head and neck cancer than females: a 13.9-fold higher risk of laryngeal cancer, a 10.9-fold higher risk of hypopharyngeal cancer, a 4.4-fold higher risk of oropharyngeal cancer, a 2.9-fold higher risk for nasopharyngeal cancer, a 1.8-fold higher risk for nasal sinus cancer, and a 1.6-fold higher risk for oral cancer. Interestingly, the HRs for head and neck cancers in males tended to increase in anatomical order closer to the esophagus in the upper aerodigestive tract: larynx > hypopharynx > oropharynx > nasopharynx > sinonasal region > mouth (Figure 4). Only the salivary gland cancer incidence did not differ between the sexes; the gland is not in the upper aerodigestive tract. *Helicobacter pylori (H. pylori)* infection causes duodenal and gastric ulcers [40] and can migrate to the upper aerodigestive tract [41]; it is found in dental plaque, saliva, tonsils, and even adenoidal tissue (associated with gastroesophageal reflux). These materials/tissues may serve as bacterial reservoirs [42]. In a comparative analysis of 109,360 patients with peptic ulcer disease and 218,720 controls detailed in the Taiwan National Health Insurance Research Database (1997–2013), Lu et al. [43] found that those with peptic ulcers were at a higher risk of laryngeal and hypopharyngeal cancers (adjusted HRs: 2.27 [95% CI: 1.16–4.44] and 2.00 [95% CI, 1.13–3.55]). In a systematic review of 244 studies, Ibrahim et al. found that the male sex was associated with a higher incidence of *H. pylori* infection. Although it remains unclear whether such infection is a significant risk factor for head and neck cancer, and how sex might affect the acquisition and persistence of the infection, our results suggest a direction for future research on head and neck cancer epidemiology.

The limitations of our study are that epidemiological data on human papillomavirus were not included and the effect of cervical cancer vaccination in females was not considered, although HPV infection increases the risk of head and neck cancers.

Laryngeal cancer is the most male-prevalent head-and-neck cancer. Concerning differences in the results of the carcinogenic effects of tobacco smoking and drinking, we investigated the prevalence of laryngeal cancer in both sexes lacking exposure to these carcinogens, but found that the incidence of laryngeal cancer remained 16.9-times higher in males. Hormone control of cancer progression might be a mechanism, as the larynx is a secondary sex organ that undergoes physiological changes during puberty, which suggests a relationship with sex hormone receptors, such as estrogen receptor. Estrogen receptors are found in head and neck subsites, especially in the larynx, and several authors have suggested that estrogens play a direct role in regulating laryngeal cancer progression [44,45,46]. Verma et al. reported that laryngeal cancers responded to 17β-estradiol via the estrogen receptor and that higher estrogen-receptor expression was correlated with a better survival [45,47]. Atef et al. found that estrogen, progesterone, and androgen receptors were positive in 56%, 50%, and 64% of 50 laryngeal cancer patients, respectively; also, the expressions of estrogen and progesterone receptors were significantly higher, whereas that of androgen receptor was lower, in patients with aggressive clinical and pathological manifestations [48]. With the possible susceptibility of laryngeal cancer to estrogens, further investigation is needed to demonstrate the effects of estrogen and estrogen receptors on laryngeal cancer progression and possible prognostic markers in the treatment of laryngeal cancer.

## 5. Conclusions

Males are much more susceptible to head and neck cancers than females regardless of whether they drink alcohol or smoke tobacco. Sex differences in the incidence of head and neck cancer are most evident in the 60 s in the lower part of the upper aerodigestive tract, such as the larynx and hypopharynx. Further research is needed.

## Figures and Tables

**Figure 1 cancers-14-02521-f001:**
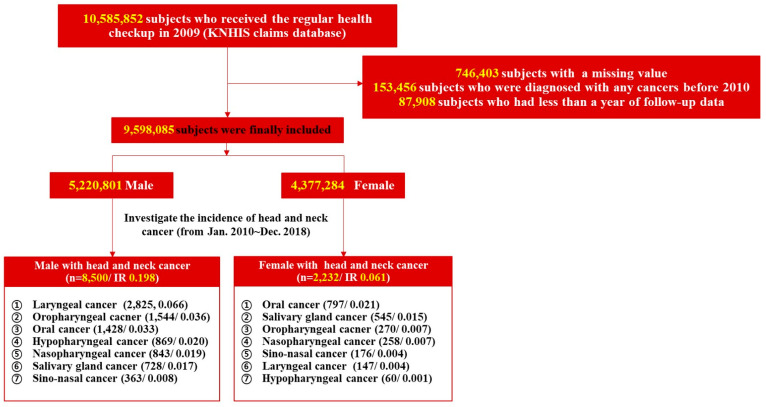
Flow chart of the study. Abbreviations: KNHIS, Korean National Health Insurance Service; IR, incidence rate per 1000 person-years.

**Figure 2 cancers-14-02521-f002:**
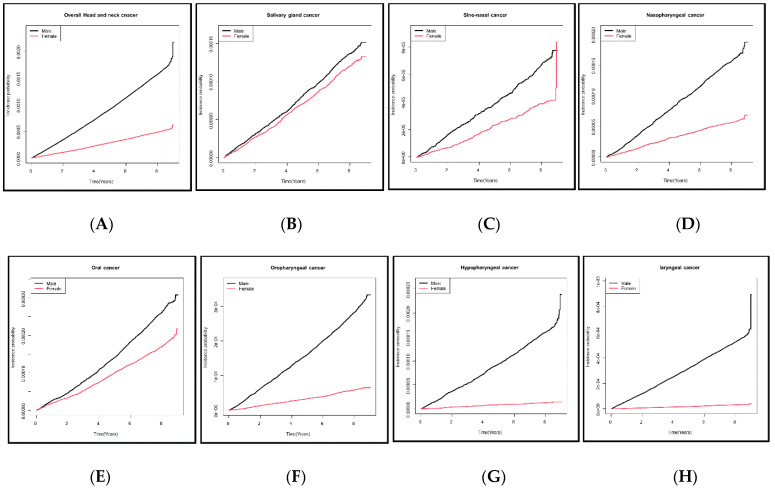
Kaplan–Meier curve of the cumulative incidence of various head and neck cancers in the original cohort of healthy males and females: (**A**) all head and neck cancers; (**B**) salivary gland cancer, (**C**) sinonasal cancer, (**D**) nasopharyngeal cancer, (**E**) oral cancer, (**F**) oropharyngeal cancer, (**G**) hypopharyngeal cancer, and (**H**) laryngeal cancer.

**Figure 3 cancers-14-02521-f003:**
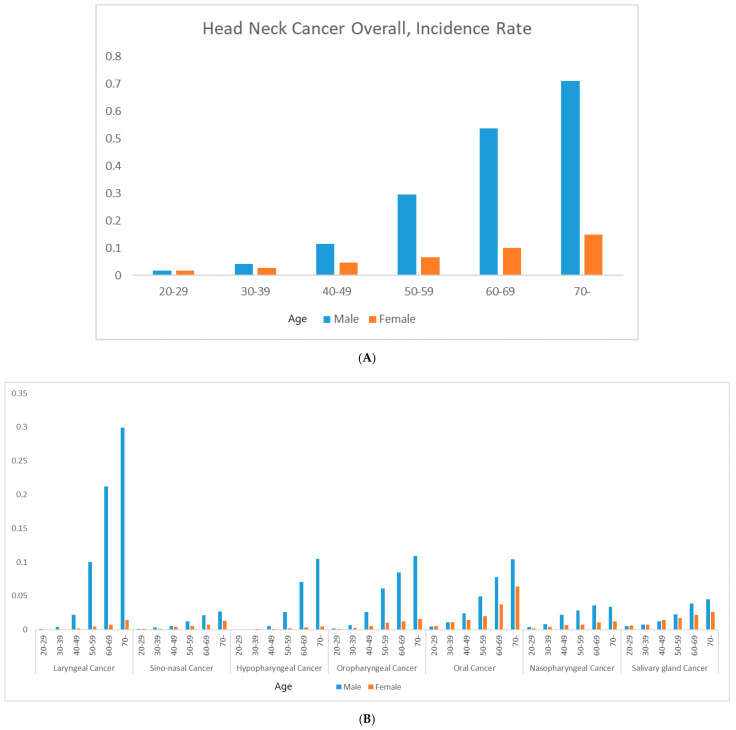
The incidence of head and neck cancers in healthy men and women by age during the 10-year follow-up period. (**A**) All head and neck cancers and (**B**) cancers by the subsites.

**Figure 4 cancers-14-02521-f004:**
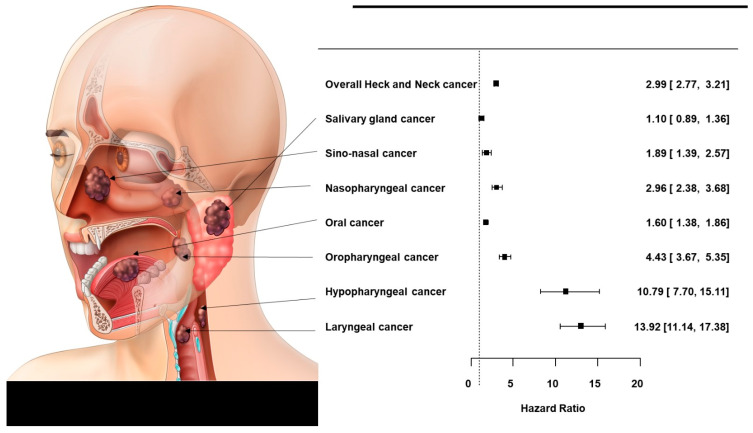
Forest plot of the hazard ratios for all subsites of head and neck cancers in males compared to females (never-smokers and never-drinkers only). A multivariate Cox’s proportional hazard model adjusted for age, body mass index, low income, regular exercise, and diabetes mellitus and hypertension status was employed.

**Table 1 cancers-14-02521-t001:** Baseline characteristics of the original cohort of subjects.

Parameters	Male (*n* = 5,220,801)	Female (*n* = 4,377,284)	*p*-Value
Age (years)	45.63 ± 13.43	48.64 ± 14.5	<0.0001 *
Age group			<0.0001 *
<40 years	1,944,807 (37.25%)	1,085,391 (24.8%)	
40–64 years	2,723,688 (52.17%)	2,619,380 (59.84%)	
≥65 years	552,306 (10.58%)	672,513 (15.36%)	
Smoking status			<0.0001 *
Never-smoker	1,612,028 (30.88%)	4,160,334 (95.04%)	
Ex-smoker	1,256,220 (24.06%)	69,232 (1.58%)	
Current smoker	2,352,553 (45.06%)	147,718 (3.37%)	
Drinking status			<0.0001 *
None	1,674,378 (32.07%)	3,264,933 (74.59%)	
Mild (<30 g/day)	2,828,959 (54.19%)	1,063,614 (24.3%)	
Heavy (≥30 g/day)	717,464 (13.74%)	48,737 (1.11%)	
Low income	785,104 (15.04%)	1,092,812 (24.97%)	<0.0001 *
Regular exercise	1,030,226 (19.73%)	678,186 (15.49%)	<0.0001 *
Height (cm)	170.01 ± 6.44	156.48 ± 6.22	<0.0001 *
Weight (kg)	69.87 ± 10.54	56.73 ± 8.42	<0.0001 *
Body mass index (kg/m^2^)	24.13 ± 3.33	23.19 ± 3.54	<0.0001 *
Body mass index (kg/m^2^)			<0.0001 *
<25	3,276,855 (62.77%)	3,189,643 (72.87%)	
≥25	1,943,946 (37.23%)	1,187,641 (27.13%)	
Waist circumference (cm)	83.56 ± 8.26	76.2 ± 9.3	<0.0001 *
Abdominal obesity(waist circumference, M ≥ 90 cm/F ≥ 85 cm)	1,115,781 (21.37%)	768,973 (17.57%)	<0.0001 *
Glucose (mM)	99.06 ± 25.77	95.07 ± 21.14	<0.0001 *
Diabetes mellitus	508,895 (9.75%)	321,744 (7.35%)	<0.0001 *
Systolic BP (mmHg)	124.66 ± 14.13	119.76 ± 15.71	<0.0001 *
Diastolic BP (mmHg)	78.05 ± 9.72	74.25 ± 10.09	<0.0001 *
Hypertension	1,395,431 (26.73%)	1,071,413 (24.48%)	<0.0001 *
eGFR (mL/min/1.73 m^2^)	87.99 ± 51.5	87.21 ± 36.52	<0.0001 *
Chronic kidney disease (eGFR < 60)	314,306 (6.02%)	344,244 (7.86%)	<0.0001 *
Total cholesterol (mM)	194.67 ± 40.93	196.15 ± 42.08	<0.0001 *
HDL cholesterol (mM)	53.56 ± 31.54	60 ± 33.83	<0.0001 *
LDL cholesterol (mM)	118.42 ± 190.63	122.75 ± 214.43	<0.0001 *
Triglyceride (mM)	129.15 (129.09–129.21)	95.75 (95.7–95.8)	<0.0001 *
Dyslipidemia	866,675 (16.6%)	869,691 (19.87%)	<0.0001 *

* Significant at *p* < 0.05. Abbreviations: BP, blood pressure; eGFR, estimated glomerular filtration rate; HDL, high-density lipoprotein; LDL, low-density lipoprotein.

**Table 2 cancers-14-02521-t002:** Multivariate Cox’s proportional hazard model for the incidence of head and neck cancer by sex. Model 1: unadjusted. Model 2: adjusted for age, body mass index (BMI), low income, smoking status, drinking status, and regular exercise. Model 3: adjusted for age, body mass index (BMI), low income, smoking status, drinking status, regular exercise, and diabetes mellitus and hypertension status.

				Per 1000 Person-Years						
	Number	Event	Duration	Rate	HR (95% CI)					
					Model 1	*p*-Value	Model 2	*p*-Value	Model 3	*p*-Value
All head and neck cancers										
Male	5,220,801	8500	42,731,332.51	0.19892	3.238 (3.09, 3.392)	<0.0001	2.83 (2.67, 3)	<0.0001	2.816 (2.656, 2.985)	<0.0001
Female	4,377,284	2232	36,231,562.63	0.0616	1 (Ref.)		1 (Ref.)		1 (Ref.)	
Laryngeal cancer										
Male	5,220,801	2825	42,747,447.1	0.06609	16.306 (13.816, 19.244)	<0.0001	11.036 (9.218, 13.213)	<0.0001	10.981 (9.171, 13.148)	<0.0001
Female	4,377,284	147	36,238,096.36	0.00406	1 (Ref.)		1 (Ref.)		1 (Ref.)	
Sinonasal cancer										
Male	5,220,801	363	42,755,919.66	0.00849	1.754 (1.465, 2.1)	<0.0001	1.77 (1.398, 2.242)	<0.0001	1.758 (1.388, 2.226)	<0.0001
Female	4,377,284	176	36,238,089.5	0.00486	1 (Ref.)		1 (Ref.)		1 (Ref.)	
Hypopharyngeal cancer										
Male	5,220,801	869	42,754,938.24	0.02033	12.345 (9.504, 16.037)	<0.0001	9.987 (7.505, 13.289)	<0.0001	9.949 (7.476, 13.239)	<0.0001
Female	4,377,284	60	36,238,435.42	0.00166	1 (Ref.)		1 (Ref.)		1 (Ref.)	
Oropharyngeal cancer										
Male	5,220,801	1544	42,752,316.34	0.03612	4.856 (4.267, 5.527)	<0.0001	4.613 (3.963, 5.371)	<0.0001	4.589 (3.942, 5.343)	<0.0001
Female	4,377,284	270	36,237,728.87	0.00745	1 (Ref.)		1 (Ref.)		1 (Ref.)	
Oral cancer										
Male	5,220,801	1428	42,752,896.93	0.0334	1.523 (1.396, 1.661)	<0.0001	1.483 (1.32, 1.666)	<0.0001	1.472 (1.31, 1.654)	<0.0001
Female	4,377,284	797	36,236,352.3	0.02199	1 (Ref.)		1 (Ref.)		1 (Ref.)	
Nasopharyngeal cancer										
Male	5,220,801	843	42,754,118.86	0.01972	2.778 (2.417, 3.194)	<0.0001	2.798 (2.353, 3.327)	<0.0001	2.805 (2.359, 3.336)	<0.0001
Female	4377284	258	36237707.17	0.00712	1 (Ref.)		1 (Ref.)		1 (Ref.)	
Salivary gland cancer										
Male	5,220,801	728	42,754,522.91	0.01703	1.132 (1.013, 1.265)	0.0285	1.145 (0.983, 1.335)	0.0815	1.139 (0.977, 1.327)	0.0966
Female	4,377,284	545	36,236,699.52	0.01504	1 (Ref.)		1 (Ref.)		1 (Ref.)	

Model 1: non-adjusted; Model 2: AGE HE_BMI INCOME_LOW SMOKING DRINKER_3LEVEL PA_REGULAR; Model 3: AGE HE_BMI INCOME_LOW SMOKING DRINKER_3LEVEL PA_REGULAR DMYN HPYN.

## Data Availability

Data available upon request due to data sharing restrictions. The data presented in this study are available upon request from the corresponding author.

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
