# Peer review of "Sex Differences in the Prevalence of Head and Neck Cancers: A 10-Year Follow-Up Study of 10 Million Healthy People"

_cancers, 2022, doi:10.3390/cancers14102521_

Round 1

Reviewer 1 Report

A difference of head and neck cancer incidence in males and females was observed decades ago. Sexual differences have been also found along various cancer sites in head neck region. When HPV infection was established as an alternative to tobacco smoking/alcohol abusing causative agent sexual dimorphism was also taken under study. So, what is a novelty in the reviewed study? In fact the one is as an extremely high number of subjects analyzed. Anyway, starting from healthy population the authors have got the same conclusion as in studies comparing simply diagnosed cancer. Again: everything is known.

Discussing a biological basis for male overrepresentation in HNC subjects the authors touch the same potential confounding factors as hormonal and immune control. Hormone- control on laryngeal function is higher than in other HN locations but is not sufficiently heighten in the text. It should be discussed together with a prevalence of laryngeal cancer compared with other cancer sites in HN. There is some explanation of difference hidden in it. Anyway this proposal has not received still a strong experimental evidence together as well as  immune related supposition. Concerning differences in results of carcinogenic effect of tobacco smoking and drinking it is worth to report the studies of laryngeal cancer in both sexes equally exposed to tobacco smoke carcinogens. It was noticed that with a recent decline of smoking the subgroup of heavily dependent on smoking was almost the same in both groups but cancer incidences still remains ca. 3 times higher in males. To close this remark I have to admit that the authors did on introduce own explanation.

The minor remark is dedicated rather to editorial staff. Citation in the text are given not in one way:. Compare lines 47, 48, 60 and 71. Go one way.

Altogether, I am somehow disappointed the reviewed study. Nevertheless, a huge population taken under study is a real aid boat for the manuscript and I suggest to consider my remarks to make the text better and acceptable.

Author Response

Dear Editor,

Thank you very much for your letter regarding our manuscript “Sex differences in the prevalence of head and neck cancers: A 10-year follow-up study of 10 million healthy people.” We appreciate the reviewers’ comments and are encouraged by their positive feedback. Below, we have provided a point-by-point response to the comments.

[EDITOR'S comments]

A difference of head and neck cancer incidence in males and females was observed decades ago. Sexual differences have been also found along various cancer sites in head neck region. When HPV infection was established as an alternative to tobacco smoking/alcohol abusing causative agent sexual dimorphism was also taken under study. So, what is a novelty in the reviewed study? In fact the one is as an extremely high number of subjects analyzed. Anyway, starting from healthy population the authors have got the same conclusion as in studies comparing simply diagnosed cancer. Again: everything is known.

Discussing a biological basis for male overrepresentation in HNC subjects the authors touch the same potential confounding factors as hormonal and immune control. Hormone- control on laryngeal function is higher than in other HN locations but is not sufficiently heighten in the text. It should be discussed together with a prevalence of laryngeal cancer compared with other cancer sites in HN. There is some explanation of difference hidden in it. Anyway this proposal has not received still a strong experimental evidence together as well as  immune related supposition. Concerning differences in results of carcinogenic effect of tobacco smoking and drinking it is worth to report the studies of laryngeal cancer in both sexes equally exposed to tobacco smoke carcinogens. It was noticed that with a recent decline of smoking the subgroup of heavily dependent on smoking was almost the same in both groups but cancer incidences still remains ca. 3 times higher in males. To close this remark I have to admit that the authors did on introduce own explanation.

The minor remark is dedicated rather to editorial staff. Citation in the text are given not in one way:. Compare lines 47, 48, 60 and 71. Go one way.

Altogether, I am somehow disappointed the reviewed study. Nevertheless, a huge population taken under study is a real aid boat for the manuscript and I suggest to consider my remarks to make the text better and acceptable.

[Answer]

Thank you for your meticulous comments. We agree fully and have added the following explanation to the Discussion.

Page 11 Line 4

Laryngeal cancer is the most male-prevalent head-and-neck cancer. Concerning differences in the results of the carcinogenic effects of tobacco smoking and drinking, we investigated the prevalence of laryngeal cancer in both sexes lacking exposure to these carcinogens, but found that the incidence of laryngeal cancer remained 16.9-times higher in males. Hormone control of cancer progression might be a mechanism, as the larynx is a secondary sex organ that undergoes physiological changes during puberty, which suggests a relationship with sex hormone receptors, such as estrogen receptor. Estrogen receptors are found in head and neck subsites, especially in the larynx, and several authors have suggested that estrogens play a direct role in regulating laryngeal cancer progression (1-3). Verma et al. reported that laryngeal cancers responded to 17β-estradiol via the estrogen receptor and that higher estrogen-receptor expression was correlated with a better survival (2, 4). Atef et al. found that estrogen, progesterone, and androgen receptors were positive in 56%, 50%, and 64% of 50 laryngeal cancer patients, respectively; also, the expressions of estrogen and progesterone receptors were significantly higher, whereas that of androgen receptor was lower, in patients with aggressive clinical and pathological manifestations (5). With the possible susceptibility of laryngeal cancer to estrogens, further investigation is needed to demonstrate the effects of estrogen and estrogen receptors on laryngeal cancer progression and possible prognostic markers in the treatment of laryngeal cancer.

Reviewer 2 Report

The authors report on a large cohort of patients from South Korea identified during one year and followed for 10 years. The data is well presented and the analysis of data adequate. Specifically, the demonstration of the independant effect of sex on the incidence of HNSCC is particularly convincing since linked to a large database.

I howerver remain with 2 main questions and elements of criticism of this study:

  1. Nasopharynx and salivary gland present a different histology, not necessarily associated with the same risk factors. They might in fact reduce the effect demonstrated for the overall population.
  2. I am surprised to see the low incidence of oropharynx cancer in SOuth Korea compared to occidental cohorts. Moreover, the presence of HPV in that population should be specifically reported to indicate if it drives or not the reduction in male:female ratio in the oropharynx subgroup.

Therefore, I would suggest that the authors report on populations with similar histology to estimate the overall risk, excluding salivary gland and nasopharynx, and only present the data of these subgroups as general information and demonstration that SCC histology is particularly linked to the male:female ratio. I would also request, if available, data on P16 positivity in the oropharynx/tonsil subgroup to define the effect of this association on the male:female ratio.

Otherwise, I think the authors did great work here which is worthy of publication. Moreover, I hope this database permits the evaluation of the prognostic effect of sex on the outcome of HNSCC in this cohort and should therefore be published.

Author Response

[EDITOR'S comments]

The authors report on a large cohort of patients from South Korea identified during one year and followed for 10 years. The data is well presented and the analysis of data adequate. Specifically, the demonstration of the independant effect of sex on the incidence of HNSCC is particularly convincing since linked to a large database.

I howerver remain with 2 main questions and elements of criticism of this study:

  1. Nasopharynx and salivary gland present a different histology, not necessarily associated with the same risk factors. They might in fact reduce the effect demonstrated for the overall population.
  2. I am surprised to see the low incidence of oropharynx cancer in SOuth Korea compared to occidental cohorts. Moreover, the presence of HPV in that population should be specifically reported to indicate if it drives or not the reduction in male:female ratio in the oropharynx subgroup.

Therefore, I would suggest that the authors report on populations with similar histology to estimate the overall risk, excluding salivary gland and nasopharynx, and only present the data of these subgroups as general information and demonstration that SCC histology is particularly linked to the male:female ratio. I would also request, if available, data on P16 positivity in the oropharynx/tonsil subgroup to define the effect of this association on the male:female ratio.

Otherwise, I think the authors did great work here which is worthy of publication. Moreover, I hope this database permits the evaluation of the prognostic effect of sex on the outcome of HNSCC in this cohort and should therefore be published.

[Answer]

 Thank you for your meticulous comments. We agree, but we cannot make the suggested change for two reasons.  First, the Korean National Health Insurance Service (KNHIS) database does not include data on p16 positivity in the oropharynx/tonsil subgroup. Second, because our contract with the statistician who handles the KNHIS database has expired, additional data analysis cannot be performed. Our aim was not to show the sex differences in the prevalence of squamous cell carcinoma but rather to investigate the sex differences in the prevalence in each subsite of head-and-neck cancer regardless of histological findings. Therefore, we believe that presenting data on the nasopharynx and salivary gland subgroups is of clinical significance. We hope that you understand and accept this decision.

Once again, we greatly appreciate the reviewers’ help. The suggested ideas have strengthened our paper. We believe that the manuscript has been improved satisfactorily and hope that it will be accepted for publication in Cancers.

References

  1. Laffont S, Seillet C, Guéry J-C. Estrogen receptor-dependent regulation of dendritic cell development and function. Frontiers in Immunology. 2017;8.
  2. Verma A, Schwartz N, Cohen DJ, Boyan BD, Schwartz Z. Estrogen signaling and estrogen receptors as prognostic indicators in laryngeal cancer. Steroids. 2019;152:108498.
  3. Fei M, Zhang J, Zhou J, Xu Y, Wang J. Sex-related hormone receptor in laryngeal squamous cell carcinoma: correlation with androgen estrogen-É‘ and prolactin receptor expression and influence of prognosis. Acta Otolaryngol. 2018;138(1):66-72.
  4. Verma A, Schwartz N, Cohen DJ, Patel V, Nageris B, Bachar G, et al. Loss of estrogen receptors is associated with increased tumor aggression in laryngeal squamous cell carcinoma. Sci Rep. 2020;10(1):4227.
  5. Atef A, El-Rashidy MA, Elzayat S, Kabel AM. The prognostic value of sex hormone receptors expression in laryngeal carcinoma. Tissue Cell. 2019;57:84-9.

Round 2

Reviewer 2 Report

I understand the limitations encountered by the authors based on the comments made previoulsy. I agree with the current submission. i do not believe the paragraph on the embryological origin of larynx brings anythingto the paper. i did not request this. I leave it to the editor to decide on its value compared to the rest of the paper. 

In the conclusion, I would propose to state the HR for H&N cancer independant of alcohol and tobacco, which I believe is the most important information out of the paper.

Author Response

Dear Editor,

Thank you very much for your letter regarding our manuscript “Sex differences in the prevalence of head and neck cancers: A 10-year follow-up study of 10 million healthy people.” We appreciate the reviewers’ comments and are encouraged by their positive feedback. Below, we have provided a point-by-point response to the comments.

[EDITOR'S comments]

I understand the limitations encountered by the authors based on the comments made previoulsy. I agree with the current submission. i do not believe the paragraph on the embryological origin of larynx brings anythingto the paper. i did not request this. I leave it to the editor to decide on its value compared to the rest of the paper. 

In the conclusion, I would propose to state the HR for H&N cancer independant of alcohol and tobacco, which I believe is the most important information out of the paper.

[Answer]

Thank you for your comments. We edited as you recommended and stated the HR for H&N cancer independant of alcohol and tobacco, which I believe is the most important information out of the paper.

Page 10 Line 8

. When never-smokers/-drinkers (only) were compared, males remained at a 2.9-fold higher risk of head-and-neck cancer than females: a 13.9-fold higher risk of laryngeal cancer, a 10.9-fold higher risk of hypopharyngeal cancer, a 4.4-fold higher risk of oropharyngeal cancer, a 2.9-fold higher risk for nasopharyngeal cancer, a 1.8-fold higher risk for nasal sinus cancer, and a 1.6-fold higher risk for oral cancer.

Once again, we greatly appreciate the reviewers’ help. The suggested ideas have strengthened our paper. We believe that the manuscript has been improved satisfactorily and hope that it will be accepted for publication in Cancers.